# Estimating the Conformal Prediction Threshold from Noisy Labels

## Abstract

Conformal Prediction (CP) is a method to control prediction uncertainty by producing a small prediction set, ensuring a predetermined probability that the true class lies within this set. This is commonly done by defining a score, based on the model predictions, and setting a threshold on this score using a validation set. In this study, we address the problem of CP calibration when we only have access to a validation set with noisy labels. We show how we can estimate the noise-free conformal threshold based on the noisy labeled data. Our solution is flexible and can accommodate various modeling assumptions regarding the label contamination process, without needing any information about the underlying data distribution or the internal mechanisms of the machine learning classifier. We develop a coverage guarantee for uniform noise that is effective even in tasks with a large number of classes. We dub our approach Noise-Aware Conformal Prediction (NACP) and show on several natural and medical image classification datasets, including ImageNet, that it significantly outperforms current noisy label methods and achieves results comparable to those obtained with a clean validation set.

## 1 Introduction

In machine learning for safety-critical applications, the model must only make predictions it is confident about. One way to achieve this is by returning a (hopefully small) set of possible class candidates that contain the true class with a predefined level of certainty. This is a natural approach for medical imaging, where safety is of the utmost importance and a human makes the final decision. This allows us to aid the practitioner, by reducing the number of possible diagnoses he needs to consider, with a controlled chance of mistake. The general approach to return a prediction set without any assumptions on the data distribution (besides i.i.d. samples) is called Conformal Prediction (CP) (Angelopoulos et al., 2023; Vovk et al., 2005). It creates a prediction set with the guarantee that the probability of the correct class being within this set meets or exceeds a specified confidence threshold. The goal is to return the smallest set possible while maintaining the confidence level guarantees. Recently, with the growing use of neural network systems in safety-critical applications such as medical imaging, CP has become an important calibration tool (Lu et al., 2022a;b; Olsson et al., 2022). We note that CP is a general framework rather than a specific algorithm. The most common approach builds the prediction set using a conformity score, and different algorithms mostly vary in terms of how the conformity score is defined.

When dealing with conformal predictions, a critical challenge arises in applications such as medical imaging due to label noise. In these domains, datasets frequently contain noisy labels stemming from ambiguous data that can confuse even clinical experts. Furthermore, physicians may disagree on the diagnosis for the same medical image, leading to inconsistencies in the ground truth labeling. Noisy labels also occur when applying differential privacy techniques to overcome privacy issues (Ghazi et al., 2021). While significant efforts have been devoted to the problem of noise-robust network training (Song et al., 2022; Xue et al., 2022), the challenge of calibrating the models has only recently begun to receive attention.

In this study, we tackle the challenge of applying CP to classification networks using a validation set with noisy labels. Einbinder et al. (2022) suggested ignoring label noise and simply applying the standard CP algorithm on the noisy labeled validation set. This strategy results in large prediction sets especially when there are many classes. A recent study suggests estimating the noise-free conformal score given its noisy version and then applying the standard CP algorithm (Penso & Goldberger, 2024). The most related study to ours is (Sesia et al., 2023) which presents a noisy CP algorithm using conservative coverage guarantee bounds which can result in large prediction sets. Here, we present a novel algorithm for CP on noisy data that yields an effective coverage guarantee even in tasks with a large number of classes. We applied the algorithm to several standard medical and scenery imaging classification datasets and show that

our method outperformed previous methods by a significant margin and achieved results comparable to those obtained by using a clean validation set.

## 2 BACKGROUND

### 2.1 CONFORMAL PREDICTION

Consider a setup involving a classification network that categorizes an input $x$ into $k$ predetermined classes. Given a coverage level of $1 - \alpha$, we aim to identify the smallest possible prediction set (a subset of these classes) ensuring the correct class is within the set with a probability of at least $1 - \alpha$. A straightforward strategy to achieve this objective involves sequentially incorporating classes from the highest to the lowest probabilities until their cumulative sum exceeds the threshold of $1 - \alpha$. Despite the network's output adopting a mathematical distribution format, it does not inherently reflect the actual class distribution. Typically, the network will not be calibrated and it tends to be overly optimistic (Guo et al., 2017). Consequently, this straightforward approach doesn't assure the inclusion of the correct class with the desired probability.

The first step of the CP algorithm involves forming a conformity score $S(x, y)$ that measures the network's uncertainty between $x$ and its true label $y$ (larger scores indicate worse agreement). The Homogeneous Prediction Sets (HPS) score (Vovk et al., 2005) is $S_{\text{HPS}}(x, y) = 1 - p(y|x; \theta)$, s.t. $\theta$ is the network parameter set. The Adaptive Prediction Score (APS) (Romano et al., 2020) is the sum of all class probabilities that are not lower than the probability of the true class:

$$S_{APS}(x, y) = \sum_{\{i|p_i \geq p_y\}} p_i, \tag{1}$$

such that $p_i = p(y = i|x; \theta)$ and $p_y$ is the probability of the label $y$. The RAPS score (Angelopoulos et al., 2021) is a variant of APS, which is defined as follows:

$$S_{RAPS}(x, y) = \sum_{\{i|p_i \geq p_y\}} p_i + a \cdot \max(0, (NC - b)) \tag{2}$$

s.t. $NC = |\{i|p_i \geq p_y\}|$ and $a, b$ are parameters that need to be tuned. RAPS is especially effective in the case of a large number of classes where it explicitly encourages small prediction sets.

We can also define a randomized version of a conformity score. For example in the case of APS we define:

$$S_{rand-APS}(x, y, u) = \sum_{\{i|p_i > p_y\}} p_i + u \cdot p_y, \qquad u \sim U[0, 1]. \tag{3}$$

The random version tends to yield the required coverage more precisely and thus it produces smaller prediction sets (Angelopoulos et al., 2023). The CP prediction set of a data point $x$ is defined as $C_q(x) = \{y|S(x, y) \leq q\}$ where $q$ is a threshold that is found using a labeled validation set $(x_1, y_1), ..., (x_n, y_n)$. The CP theorem states that if we set $q$ to be the $(1 - \alpha)$ quantile of the conformal scores $S(x_1, y_1), ..., S(x_n, y_n)$ we can guarantee that $1 - \alpha \leq p(y \in C(x)) \leq 1 - \alpha + \frac{1}{n+1}$, where $x$ is a test point and $y$ is its the unknown true label (Vovk et al., 2005). In the random case there is still a coverage guarantee, which is defined by marginalizing over all test points $x$ and samplings $u$ from the uniform distribution (Romano et al., 2020). Note that the coverage guarantee is for a marginal probability over all possible test points and coverage may be worse or better for different points. It can be proved that obtaining a conditional coverage guarantee is impossible (Foygel Barber et al., 2021).

## 3 OUR APPROACH

### 3.1 SETTING THE THRESHOLD GIVEN NOISY LABELS

Here we show how, given a simple noise model and a known noise level, we can get the correct CP threshold based on noisy data. We will generalize this beyond the simple noise model in the following section. Consider a network that classifies an input $x$ into $k$ pre-defined classes. Given a conformity score $S(x, y)$ and a specified coverage $1 - \alpha$, the goal of the conformal prediction algorithm is to find a minimal $q$ such that $p(y \in C_q(x)) \geq 1 - \alpha$. Let $(x_1, \tilde{y}_1), ..., (x_n, \tilde{y}_n)$ be a validation set with noisy labels and let $y_i$ be the unknown correct label of $x_i$. Let $s_i = S(x_i, \tilde{y}_i)$ be the conformity score of $(x_i, \tilde{y}_i)$. We assume that the label noise follows a uniform distribution, where with a probability of $\epsilon$, the correct label is replaced by a label that is randomly sampled from the $k$ classes:

$$p(\tilde{y} = j|y = i) = \mathbb{1}_{\{i=j\}}(1 - \epsilon) + \frac{\epsilon}{k}. \tag{4}$$

Uniform noise is relevant, for example, when applying differential privacy techniques to overcome privacy issues (Ghazi et al., 2021). In that setup the noise level $\epsilon$ is usually known. In other applications such as medical imaging, where the noise parameter $\epsilon$ is not given, it can be estimated with sufficient accuracy from the noisy-label data during training (Zhang et al., 2021; Li et al., 2021; Lin et al., 2023). We can write $\tilde{y}$ as $\tilde{y} = (1 - z) \cdot y + z \cdot u$, s.t. $u$ is a random label uniformly sampled from $\{1, ..., k\}$ and $z$ is a binary random variable ($p(z = 1) = \epsilon$) indicating whether the label of the sample $(x, y)$ was replaced by a random label or not. For each candidate threshold $q$ denote:

$$F^c(q) = p(y \in C_q(x)), \qquad F^n(q) = p(\tilde{y} \in C_q(x)), \qquad F^r(q) = p(u \in C_q(x)),$$

where $F^c$, $F^n$, and $F^r$ represent the clean, noisy and random labels. Note as well that each one is the CDF of the appropriate conformal score function, e.g., $F^c(q) = p(y \in C_q(x)) = p(S(x, y) \leq q)$.

It is easily verified that

$$F^n(q) = p(z = 0)F^c(q) + p(z = 1)F^r(q) = (1 - \epsilon)F^c(q) + \epsilon F^r(q). \tag{5}$$

For each value $q$, we can estimate $F^n(q)$ from the noisy validation set:

$$\hat{F}^n(q) = \frac{1}{n} \sum_i \mathbb{1}_{\{\tilde{y}_i \in C_q(x_i)\}} = \frac{1}{n} \sum_i \mathbb{1}_{\{s_i \leq q\}}. \tag{6}$$

Note that $q$ is the $\hat{F}^n(q)$-quantile of $s_1, ..., s_n$. Similarly we can also estimate $F^r(q)$:

$$\hat{F}^r(q) = \frac{1}{n} \sum_i p(u_i \in C_q(x_i)) = \frac{1}{n} \sum_i \frac{|C_q(x_i)|}{k}, \tag{7}$$

s.t. $u_i$ is uniformly sampled from $\{1, ..., k\}$.

By substituting (6) and (7) in (5) we obtain an estimation of $F^c(q) = p(y \in C_q(x))$ based on the noisy validation set and the noise level $\epsilon$:

$$\hat{F}^c(q) = \frac{\hat{F}^n(q) - \epsilon \hat{F}^r(q)}{1 - \epsilon}. \tag{8}$$

For each candidate $q$ we first compute $\hat{F}^n(q)$ and $\hat{F}^r(q)$ and then by using (8) obtain the coverage estimation $\hat{F}^c(q)$. Given a coverage requirement $(1 - \alpha)$, we can thus use the noisy validation set to find a threshold $q$ such that $\hat{F}^c(q) = 1 - \alpha$. Note that since $F^c(q)$ is monotonous, it seems reasonable to search for the threshold $q$ using the bisection method. However, as $\hat{F}^c(q)$ is an approximation based on the *difference* between two monotonic functions, it might not be exactly monotonous. We therefore find the threshold $q$ using an exhaustive grid search. We note that even with an exhaustive search the entire runtime is negligible compared to the training time. Furthermore, we can narrow the threshold search domain as follows:

**Lemma 3.1.** *For every threshold $q$ we have: $\hat{F}^n_q/k \leq \hat{F}^r(q)$.*

*Proof.* Denote $A = \{i | \hat{y}_i \in C_q(x_i)\}$ and $B = \{i | 1 \leq |C_q(x_i)|\}$. Note that $\hat{F}^n(q) = |A|/n$.

$$|B| = \sum_{i \in B} 1 \leq \sum_{i \in B} |C_q(x_i)| \leq \sum_{i=1}^n |C_q(x_i)| = nk\hat{F}^r(q).$$

Finally $A \subset B$ implies that: $\hat{F}^n(q) = |A|/n \leq |B|/n \leq k\hat{F}^r(q)$. $\qquad\square$

**Theorem 3.2.** *Let $q_1$ be the $(1 - \alpha)(1 - \epsilon)/(1 - \frac{\epsilon}{k})$ quantile of $s_1, ..., s_n$ and let $q_2$ be the $(1 - \alpha) + \alpha\epsilon$ quantile. If $q$ satisfies $\hat{F}^c(q) = 1 - \alpha$ then $q_1 \leq q \leq q_2$.*

*Proof.* Assume $q$ satisfies $\hat{F}^c(q) = 1 - \alpha$. Eq. (8) implies that

$$1 - \alpha = \hat{F}^c(q) = \frac{\hat{F}^n(q) - \epsilon \hat{F}^r(q)}{1 - \epsilon} \Rightarrow \hat{F}^n(q) = (1 - \alpha)(1 - \epsilon) + \epsilon \hat{F}^r(q). \tag{9}$$

Since $0 \leq \hat{F}^r(q) \leq 1$ we get that:

$$(1 - \alpha)(1 - \epsilon) \leq \hat{F}^n(q) \leq (1 - \alpha) + \alpha\epsilon = \hat{F}^n(q_2). \tag{10}$$

---

**Algorithm 1** Noise-Aware Conformal Prediction (NACP) for uniform noise

---

1: Input: A conformity score $S(x, y)$, a coverage level $1 - \alpha$ and a validation set $(x_1, \tilde{y}_1), ..., (x_n, \tilde{y}_n)$, s.t. the labels are corrupted by a uniform noise with parameter $\epsilon$.
2: Set $q_1$ to be the $(1-\alpha)(1-\epsilon)/(1 - \frac{\epsilon}{k})$ quantile of $S(x_1, \tilde{y}_1), ..., S(x_n, \tilde{y}_n)$ and set $q_2$ to be $((1-\alpha) + \alpha\epsilon)$ quantile.
3: For each candidate threshold $q$ compute:

$$\hat{F}^n(q) = \frac{1}{n} \sum_i \mathbb{1}_{\{\tilde{y}_i \in C_q(x_i)\}}, \qquad \hat{F}^r(q) = \frac{1}{n} \sum_i \frac{|C_q(x_i)|}{k}, \qquad \hat{F}^c(q) = \frac{\hat{F}^n(q) - \epsilon\hat{F}^r(q)}{1 - \epsilon}$$

4: Apply a grid search to find $q \in [q_1, q_2]$ that satisfies $\hat{F}^c(q) = 1 - \alpha$.
5: The prediction set of a test sample $x$ is $C_q(x) = \{y \mid S(x, y) < q\}$.
6: Coverage guarantee: $p(y \in C_q(x)) \geq 1 - \alpha - \Delta(n, \epsilon, \delta)$ with probability $(1 - \delta)$ over the noisy validation set sampling (see Theorem 3.5).

---

For every $q$ we have $\hat{F}^n(q)/k \leq \hat{F}^r(q)$ (Lemma 3.1). Hence, $(1-\alpha)(1-\epsilon) \leq \hat{F}^n(q)$ (10) implies that $(1-\alpha)(1-\epsilon)/k \leq \hat{F}^r(q)$. Combining this inequality with Eq. (9) yields a better lower bound: $(1-\alpha)(1-\epsilon)(1+\epsilon/k) \leq \hat{F}^n(q)$. Iterating this process yields:

$$(1-\alpha)(1-\epsilon)\left(1 + \frac{\epsilon}{k} + \left(\frac{\epsilon}{k}\right)^2 + \dots\right) = (1-\alpha)\frac{1-\epsilon}{1 - \frac{\epsilon}{k}} = \hat{F}^n(q_1) \leq \hat{F}^n(q).$$

Finaly, $\hat{F}^n(q)$ is a monotonically increasing function of $q$ which implies that $q_1 \leq q \leq q_2$. $\qquad\square$

As an alternative to the grid search we can sort the noisy conformity scores $s_i = S(x_i, \tilde{y}_i)$ and look for the minimal $i$ such that $\hat{F}^c(s_i) \geq 1 - \alpha$. In the noise-free case $\hat{F}^c$ is piece-wise constant, with jumps determined exactly by the order statistics $s_i$, namely, $\hat{F}^c(s_i) = i/n$ and thus this algorithm coincides with the standard CP algorithm. In the noisy case $\hat{F}^c(q)$ depends on the conformity scores of all the $k$ classes and thus its structure is more complicated. We dub our algorithm Noise-Aware Conformal Prediction (NACP), and summarize it in Algorithm Box 1. Note that in the noise-free case ($\epsilon = 0$) the NACP algorithm coincides with the standard CP algorithm and selects $q$ that satisfies $\hat{F}^c(q) = \hat{F}^n(q) = 1 - \alpha$, i.e., $q$ is the $1 - \alpha$ quantile of the validation set conformity scores.

### 3.2 PREDICTION SIZE COMPARISON

We next compare our NACP approach analytically to Noisy-CP (Einbinder et al., 2022) in terms of the average size of the prediction set.

**Theorem 3.3.** *Let $q$ and $\tilde{q}$ be the thresholds computed by the NACP and the Noisy-CP algorithms respectively. Then $q \leq \tilde{q}$ if and only if $\hat{F}^r(\tilde{q}) \leq (1-\alpha)$.*

*Proof.* The threshold $\tilde{q}$ computed by the Noisy-CP algorithm (by applying standard CP on the noisy validations set) satisfies $\hat{F}^n(\tilde{q}) = (1-\alpha)$. The true threshold $q$ satisfies $\hat{F}^n(q) = (1-\alpha)(1-\epsilon) + \epsilon\hat{F}^r(q)$ (9). Looking at the difference

$$\hat{F}^n(\tilde{q}) - \hat{F}^n(q) = 1 - \alpha - (1-\alpha)(1-\epsilon) - \epsilon\hat{F}^r(q) = \epsilon(1 - \alpha - \hat{F}^r(q)). \tag{11}$$

Hence from the monotonicity of $\hat{F}^n(q)$ we have $q \leq \tilde{q}$ iff $\hat{F}^n(q) \leq \hat{F}^n(\tilde{q})$ iff $\hat{F}^r(q) \leq 1 - \alpha$. $\qquad\square$

The theorem above states that if the size of the prediction set obtained by NACP is less than $k(1 - \alpha)$, NACP is more effective than Noisy-CP. For example, assume $k = 100$ and $1 - \alpha = 0.9$. In this case, if the average size of the NACP prediction set is less than 90, NACP is more effective than Noisy-CP. We also see from eq. (11) that the smaller $\hat{F}^r$ is the larger the gap between the two methods. Since $\hat{F}^r$ is inversely proportional to the number of classes, we expect the difference to be substantial when there is a large number of classes to consider, which is exactly where CPs' ability to reliably exclude possible classes is very useful. In our experiments, we indeed found a considerable gap between the two methods when we experimented on classification tasks with a large number of classes.

### 3.3 COVERAGE GUARANTEES

We next provide a coverage guarantee for NACP. We show that if we apply the NACP to find a threshold $q$ for $1 - \alpha + \Delta$, then $P(y \in C_q(x)) \geq 1 - \alpha$ were $\Delta$ depends on the validation set size. $\Delta$ is a finite-sample term that is needed to approximate the CDF to set the threshold instead of simply picking a predefined quantile. Because $\Delta$ can be computed, one can adjust the $\alpha$ used in the NACP algorithm to get the desired coverage guarantee. However, we note that we empirically found this bound to be over-conservative, and that the un-adjusted method does reach the desired coverage.

**Lemma 3.4.** *Given $\delta > 0$, define $\Delta = \sqrt{\frac{\log(4/\delta)}{2nh^2}}$ such that $h = \frac{1-\epsilon}{1+\epsilon}$ and $n$ is the size of the noisy validation set. Then*

$$p(\sup_q |F^c(q) - \hat{F}^c(q)| > \Delta) \leq \delta, \tag{12}$$

*such that the probability is over the validation set.*

*Proof.* The Dvoretzky–Kiefer–Wolfowitz (DKW) inequality (Massart, 1990) states that if we estimate a CDF $F$ from $n$ samples using the empirical CDF $F_n$ then $p(\sup_x |F_n(x) - F(x)| > \Delta) \leq 2\exp(-2n\Delta^2)$. Eq. (8) defines $\hat{F}^c(q)$ using $\hat{F}^n(q)$ and $\hat{F}^r(q)$. Both are empirical CDF, so from the DKW theorem and the union bound we get that:

$$p(\sup_q |F^r(q) - \hat{F}^r(q)| > h\Delta \text{ or } \sup_q |F^n(q) - \hat{F}^n(q)| > h\Delta) \leq 4\exp(-2nh^2\Delta^2) = \delta. \tag{13}$$

Using eq. (8) we get that with probability at least $1 - \delta$ for every $q$:

$$\hat{F}^c(q) = \frac{\hat{F}^n(q) - \epsilon\hat{F}^r(q)}{1 - \epsilon} \leq \frac{(F^n(q) + h\Delta) - \epsilon(F^r(q) - h\Delta)}{1 - \epsilon}$$

$$= F^c(q) + \frac{h\Delta + \epsilon h\Delta}{1 - \epsilon} = F^c(q) + h\Delta\frac{1 + \epsilon}{1 - \epsilon} = F^c(q) + \Delta. \tag{14}$$

Similarly, we can show that $\hat{F}^c(q) \geq F^c(q) - \Delta$ which completes the proof. $\square$

The proof of the main theorem now follows the standard CP proof, taking the inaccuracy in estimating $F^c(q)$ into account.

**Theorem 3.5.** *Assume you have a noisy validation set of size $n$ with noise level $\epsilon$ and set $\Delta(n, \epsilon, \delta) = \sqrt{\frac{\log(4/\delta)}{2nh^2}}$ s.t. $h = \frac{1-\epsilon}{1+\epsilon}$ and that you pick $q$ such that $\hat{F}^c(q) = 1 - \alpha + \Delta$. Then with probability at least $1 - \delta$ (over the validation set), we have that if $(x, y)$ are sampled from the clear label distribution we get:*

$$1 - \alpha \leq p(y \in C_q(x)) \leq 1 - \alpha + 2\Delta.$$

*Proof.* Given a clean test pair $(x, y)$, with probability $\delta$ over the validation set, we have:

$$p(y \in C_q(x)) = p(S(x, y) < q) = F^c(q) \geq \hat{F}^c(q) - \Delta = 1 - \alpha.$$

In a similar way: $p(y \in C_q(x)) = F^c(q) \leq \hat{F}^c(q) + \Delta = 1 - \alpha + 2\Delta.$ $\square$

As the size of the noisy validation set, $n$, tends to infinity, $\Delta$ converges to zero and thus the noisy threshold converges to the noise-free threshold.

Sesia et al. (2023) proposed a CP algorithm for the same setup of noisy labels where the noise matrix is given (or estimated based on noise-free data). They used the distribution of correct labels given the noisy labels while we use the more natural distribution of the noisy labels given the correct label. As a result, they need to know the class frequencies for both the clean and noisy labels, whereas we do not. Another major difference between the algorithms is the finite sample coverage guarantee term. In Section 4 we empirically compare the finite sample terms of the two methods and show that while ours is effective for the case of uniform noise, their approach fails on datasets with a large number of classes such as CIFAR-100, TinyImageNet, and ImageNet, and produces prediction sets that contain all classes. A further distinction is that their finite sample coverage guarantee is established for the average of all the noisy validation sets. In contrast, our approach provides an individual coverage guarantee for nearly all $(1 - \delta)$ of the sampled noisy validation sets.

---

**Algorithm 2** Noise-Aware Conformal Prediction (NACP) for a noise matrix model

---

1: Input: A conformity score $S(x, y)$, a coverage level $1-\alpha$ and a validation set $(x_1, \tilde{y}_1), ..., (x_n, \tilde{y}_n)$, s.t. the labels are corrupted by a noise matrix $P$.
2: For each candidate threshold $q$ compute:

$$\hat{M}_q(\ell, i) = \frac{1}{n} \sum_j \mathbb{1}_{\{\tilde{y}_j = i, \ \ell \in C_q(x_j)\}}, \quad i, \ell = 1, .., k.$$

$$\hat{F}^c(q) = \text{Tr}(\hat{M}_q P^{-1}).$$

3: Apply a grid search to find $q$ that satisfies $\hat{F}^c(q) = 1-\alpha$.
4: The prediction set of a test sample $x$ is $C_q(x) = \{y \mid S(x, y) < q\}$.

---

### 3.4 A MORE GENERAL NOISE MODEL

Next, we will extend our approach to a more general noise model. We will assume that the noisy label $\tilde{y}$ is independent of $x$ given $y$. We also assume that the noise matrix $P(i, j) = p(\tilde{y} = j | y = i)$ is known and that the matrix P is invertible. For each $q$ define the following matrices for the clear and the noisy data: $M_q^c(\ell, i) = p(\ell \in C_q(x), y = i)$ and $M_q(\ell, i) = p(\ell \in C_q(x), \tilde{y} = i)$. Assuming that, given the true label $y$, the r.v. $x$ and $\tilde{y}$ are independent, we obtain:

$$M_q(\ell, i) = p(\ell \in C_q(x), \tilde{y} = i) = \sum_j p(\ell \in C_q(x), \tilde{y} = i, y = j) \tag{15}$$

$$= \sum_j p(\ell \in C_q(x), y = j) p(\tilde{y} = i | y = j) = \sum_j M_q^c(\ell, j) P(j, i).$$

We can write (15) in matrix notation: $M_q = M_q^c P$. Eq. (15) implies that:

$$F^c(q) = p(y \in C_q(x)) = p(y \in C_q(x)) = \sum_i p(i \in C_q(x), y = i) = \sum_i M_q^c(i, i) = \text{Tr}(M_q P^{-1}). \tag{16}$$

We can estimate matrix $M_q$ from the noisy samples:

$$\hat{M}_q(\ell, i) = \frac{1}{n} \sum_j \mathbb{1}_{\{\tilde{y}_j = i, \ \ell \in C_q(x_j)\}}, \quad i, \ell = 1, .., k. \tag{17}$$

Substituting (17) in (16) yields an estimation of the probability $F^c(q) = p(y \in C_q(x))$:

$$\hat{F}^c(q) = \text{Tr}(\hat{M}_q P^{-1}). \tag{18}$$

The final step is applying a grid search to find a threshold $q$ such that $\hat{F}^c(q) = 1 - \alpha$.

In the case that $P$ is a uniform noise matrix (4), the Sherman-Morison formula implies that $P^{-1} = (\frac{1}{1-\epsilon} I - \frac{\epsilon}{(1-\epsilon)k} \mathbf{1}\mathbf{1}^\top)$. Therefore,

$$\hat{F}^c(q) = \text{Tr}(\hat{M}_q P^{-1}) = \frac{1}{1-\epsilon} \sum_i \hat{M}_q(i, i) - \frac{\epsilon}{(1-\epsilon)k} \sum_{\ell, i} \hat{M}_q(\ell, i) = \frac{\hat{F}^n(q) - \epsilon \hat{F}^r(q)}{1 - \epsilon}.$$

Thus in the case of a uniform noise the coverage estimation (18) coincides with (8). If the noise matrix is unknown, it can be estimated from the noisy-label data during training (Zhang et al., 2021; Li et al., 2021; Lin et al., 2023). The NACP method for a noise matrix model is summarized in Algorithm 2.

We can extend the finite sample term $\Delta$ that was developed for a uniform noise to obtain a theoretical coverage guarantee for a noise matrix model (see Section A.2). However, this approach yields large prediction sets especially in tasks with many classes and thus is ineffective. In the experiment section we show that in practice, even without adding finite sample terms, we obtain the required coverage probability.

## 4 EXPERIMENTS

In this section, we evaluate the capabilities of our NACP algorithm on various medical and scenery imaging datasets.

Table 1: APS and RAPS (randomized versions) calibration results for $1-\alpha = 0.9$ and $\epsilon = 0.2$. We report the mean and the std over 1000 different splits. Bold for best result with theoretical guarantees.

| Table 1 | | rand-APS | | rand-RAPS | |
|---|---|---|---|---|---|
| Dataset | CP Method | size $\downarrow$ | coverage (%) | size $\downarrow$ | coverage (%) |
| TissueMNIST | CP (Oracle) | $2.33 \pm 0.01$ | $90.01 \pm 0.23$ | $2.09 \pm 0.01$ | $90.01 \pm 0.23$ |
| | Noisy-CP | $4.33 \pm 0.04$ | $98.98 \pm 0.07$ | $4.29 \pm 0.04$ | $99.12 \pm 0.06$ |
| | NR-CP | $3.19 \pm 0.02$ | $96.09 \pm 0.09$ | $3.22 \pm 0.02$ | $96.98 \pm 0.08$ |
| | ACNL | $3.07 \pm 0.04$ | $95.55 \pm 0.24$ | $2.52 \pm 0.04$ | $91.84 \pm 0.42$ |
| | NACP | $\mathbf{2.50 \pm 0.02}$ | $\mathbf{91.67 \pm 0.27}$ | $\mathbf{2.42 \pm 0.01}$ | $\mathbf{92.96 \pm 0.08}$ |
| | NACP (w/o $\Delta$) | $2.34 \pm 0.02$ | $90.05 \pm 0.27$ | $2.32 \pm 0.01$ | $92.18 \pm 0.09$ |
| OrganSMNIST | CP (Oracle) | $1.63 \pm 0.02$ | $90.0 \pm 0.58$ | $1.63 \pm 0.02$ | $90.02 \pm 0.58$ |
| | Noisy-CP | $5.62 \pm 0.17$ | $99.70 \pm 0.06$ | $5.75 \pm 0.16$ | $99.43 \pm 0.09$ |
| | NR-CP | $2.88 \pm 0.06$ | $97.95 \pm 0.14$ | $3.82 \pm 0.13$ | $98.98 \pm 0.12$ |
| | ACNL | $2.91 \pm 0.26$ | $97.92 \pm 0.53$ | $2.90 \pm 0.28$ | $97.91 \pm 0.53$ |
| | NACP | $\mathbf{1.95 \pm 0.06}$ | $\mathbf{94.03 \pm 0.61}$ | $\mathbf{1.95 \pm 0.05}$ | $\mathbf{94.04 \pm 0.59}$ |
| | NACP (w/o $\Delta$) | $1.63 \pm 0.03$ | $90.15 \pm 0.70$ | $1.63 \pm 0.03$ | $90.02 \pm 0.67$ |
| OrganCMNIST | CP (Oracle) | $1.75 \pm 0.04$ | $90.04 \pm 0.60$ | $1.19 \pm 0.01$ | $90.06 \pm 0.58$ |
| | Noisy-CP | $6.07 \pm 0.19$ | $99.96 \pm 0.02$ | $5.54 \pm 0.19$ | $99.80 \pm 0.04$ |
| | NR-CP | $2.39 \pm 0.03$ | $98.39 \pm 0.15$ | $3.26 \pm 0.99$ | $99.70 \pm 0.08$ |
| | ACNL | $2.23 \pm 0.09$ | $97.56 \pm 0.56$ | $1.62 \pm 0.07$ | $97.56 \pm 0.55$ |
| | NACP | $\mathbf{1.92 \pm 0.04}$ | $\mathbf{93.99 \pm 0.60}$ | $\mathbf{1.33 \pm 0.02}$ | $\mathbf{93.97 \pm 0.60}$ |
| | NACP (w/o $\Delta$) | $1.74 \pm 0.04$ | $90.15 \pm 0.67$ | $1.18 \pm 0.02$ | $90.12 \pm 0.69$ |
| OrganAMNIST | CP (Oracle) | $1.03 \pm 0.01$ | $90.04 \pm 0.38$ | $1.02 \pm 0.01$ | $90.02 \pm 0.37$ |
| | Noisy-CP | $5.45 \pm 0.12$ | $100.0 \pm 0.00$ | $5.45 \pm 0.12$ | $99.95 \pm 0.01$ |
| | NR-CP | $1.40 \pm 0.01$ | $98.76 \pm 0.09$ | $4.54 \pm 1.02$ | $99.94 \pm 0.02$ |
| | ACNL | $1.17 \pm 0.01$ | $95.53 \pm 0.38$ | $1.15 \pm 0.01$ | $95.53 \pm 0.39$ |
| | NACP | $\mathbf{1.08 \pm 0.01}$ | $\mathbf{92.56 \pm 0.40}$ | $\mathbf{1.07 \pm 0.01}$ | $\mathbf{92.55 \pm 0.40}$ |
| | NACP (w/o $\Delta$) | $1.03 \pm 0.01$ | $90.22 \pm 0.44$ | $1.02 \pm 0.01$ | $90.20 \pm 0.44$ |
| PathMNIST | CP (Oracle) | $1.10 \pm 0.02$ | $90.03 \pm 0.47$ | $1.03 \pm 0.01$ | $90.0 \pm 0.45$ |
| | Noisy-CP | $4.69 \pm 0.13$ | $100.00 \pm 0.00$ | $4.55 \pm 0.12$ | $99.95 \pm 0.02$ |
| | NR-CP | $1.43 \pm 0.01$ | $98.24 \pm 0.12$ | $1.53 \pm 0.03$ | $99.00 \pm 0.09$ |
| | ACNL | $1.22 \pm 0.02$ | $94.95 \pm 0.49$ | $1.30 \pm 0.04$ | $95.35 \pm 0.55$ |
| | NACP | $\mathbf{1.18 \pm 0.02}$ | $\mathbf{93.53 \pm 0.48}$ | $\mathbf{1.11 \pm 0.01}$ | $\mathbf{93.51 \pm 0.47}$ |
| | NACP (w/o $\Delta$) | $1.10 \pm 0.02$ | $90.25 \pm 0.52$ | $1.03 \pm 0.01$ | $90.24 \pm 0.53$ |

Table 2: Finite sample correction terms $\Delta$ of NACP and ACNL (Sesia et al., 2023) for several medical imaging datasets and two noise levels, $n$ is the size of the validation set.

| Dataset | $n$ | #classes | NACP | | ACNL | |
|---|---|---|---|---|---|---|
| | | | $\epsilon = 0.1$ | $\epsilon = 0.2$ | $\epsilon = 0.1$ | $\epsilon = 0.2$ |
| TissueMNIST | 35460 | 8 | $\mathbf{0.0132}$ | $\mathbf{0.0162}$ | 0.0158 | 0.0294 |
| OrganSMNIST | 5640 | 11 | $\mathbf{0.0331}$ | $\mathbf{0.0406}$ | 0.0378 | 0.0721 |
| OrganCMNIST | 5304 | 11 | $\mathbf{0.0341}$ | $\mathbf{0.0419}$ | 0.0380 | 0.0721 |
| OrganAMNIST | 12135 | 11 | $\mathbf{0.0225}$ | $\mathbf{0.0277}$ | 0.0263 | 0.0509 |
| PathMNIST | 8592 | 9 | $\mathbf{0.0268}$ | $\mathbf{0.0329}$ | 0.0426 | 0.0824 |

**Compared methods.** Our method takes an existing conformity score $S$ and computes a threshold $q$ that takes into account the label noise level. We implemented three popular conformal prediction scores, namely APS (Romano et al., 2020), RAPS (Angelopoulos et al., 2021) and HPS (Vovk et al., 2005). For each score $S$, we compared the following four CP methods: (1) CP (Oracle) - using a validation set with clean labels, (2) Noisy-CP - applying a standard CP on noisy labels without any modifications (Einbinder et al., 2022), (3) NR-CP (Penso & Goldberger, 2024) applying standard CP on an estimation of the noise-free score (4) Adaptive Conformal Classification with Noisy labels (ACNL) (Sesia et al., 2023) and (5) NACP - our approach (6) NACP (w/o $\Delta$) - our approach without the finite sample coverage guarantee $\Delta$. For methods (3) and (4) we used their official codes [1] [2]. We share our code for reproducibility[3].

**Evaluation Measures**. We evaluated each CP method based on the average size of the prediction sets (where a small value means high efficiency) and the fraction of test samples for which the prediction sets contained the ground-truth

---

[1] https://github.com/cobypenso/Noise-Robust-Conformal-Prediction

[2] https://github.com/msesia/conformal-label-noise

[3] https://anonymous.4open.science/r/Noise-Aware-Conformal-Prediction

Table 3: CP calibration results on CIFAR-10, CIFAR-100, Tiny-ImageNet and ImageNet for $1-\alpha = 0.9$ and $\epsilon = 0.2$. We report the mean and the std over 1000 different splits. Bold for best result with theoretical guarantees.

| Table 3 | | CIFAR-10 (10 classes) | | CIFAR-100 (100 classes) | | Tiny-ImageNet (200 classes) | | ImageNet (1000 classes) | |
|---|---|---|---|---|---|---|---|---|---|
| Dataset | CP Method | size ↓ | coverage(%) | size ↓ | coverage(%) | size ↓ | coverage(%) | size ↓ | coverage(%) |
| rand-APS | CP (Oracle) | $1.1 \pm 0.01$ | $90.0 \pm 0.62$ | $6.5 \pm 0.20$ | $90.0 \pm 0.43$ | $14.9 \pm 0.60$ | $90.0 \pm 0.61$ | $16.6 \pm 0.33$ | $90.0 \pm 0.26$ |
| | Noisy-CP | $5.1 \pm 1.29$ | $99.9 \pm 0.04$ | $50.5 \pm 1.29$ | $99.8 \pm 0.04$ | $99.7 \pm 3.67$ | $99.7 \pm 0.08$ | $502.6 \pm 8.56$ | $99.9 \pm 0.01$ |
| | NR-CP | $2.2 \pm 0.07$ | $98.9 \pm 0.12$ | $37.6 \pm 0.86$ | $99.6 \pm 0.06$ | $79.6 \pm 2.82$ | $99.3 \pm 0.11$ | $275.6 \pm 27.1$ | $99.7 \pm 0.06$ |
| | ACNL | $1.5 \pm 0.06$ | $96.0 \pm 0.61$ | $100.0 \pm 0.00$ | $100.0 \pm 0.00$ | $200.0 \pm 0.00$ | $100.0 \pm 0.00$ | $1000.0 \pm 0.00$ | $100.0 \pm 0.00$ |
| | NACP | $\mathbf{1.3 \pm 0.04}$ | $\mathbf{94.4 \pm 0.62}$ | $\mathbf{9.0 \pm 0.46}$ | $\mathbf{93.0 \pm 0.49}$ | $\mathbf{22.6 \pm 1.87}$ | $\mathbf{93.7 \pm 0.71}$ | $\mathbf{20.9 \pm 0.72}$ | $\mathbf{91.9 \pm 0.32}$ |
| | NACP (w/o Δ) | $1.1 \pm 0.02$ | $90.1 \pm 0.70$ | $6.4 \pm 0.28$ | $89.9 \pm 0.54$ | $14.0 \pm 0.91$ | $89.4 \pm 0.81$ | $16.7 \pm 0.51$ | $90.0 \pm 0.34$ |
| rand-RAPS | CP (Oracle) | $1.1 \pm 0.01$ | $90.0 \pm 0.61$ | $4.0 \pm 0.08$ | $90.0 \pm 0.43$ | $6.9 \pm 0.19$ | $90.0 \pm 0.62$ | $6.3 \pm 0.06$ | $90.0 \pm 0.27$ |
| | Noisy-CP | $5.1 \pm 0.18$ | $99.9 \pm 0.04$ | $50.5 \pm 1.33$ | $99.8 \pm 0.03$ | $101.4 \pm 3.58$ | $99.5 \pm 0.09$ | $501.6 \pm 8.51$ | $99.9 \pm 0.01$ |
| | NR-CP | $3.3 \pm 0.20$ | $99.6 \pm 0.08$ | $50.1 \pm 1.08$ | $99.8 \pm 0.03$ | $100.6 \pm 2.88$ | $99.5 \pm 0.09$ | $455.2 \pm 20.7$ | $99.9 \pm 0.02$ |
| | ACNL | $1.3 \pm 0.03$ | $94.6 \pm 0.65$ | $100.0 \pm 0.00$ | $100.0 \pm 0.00$ | $200.0 \pm 0.00$ | $100.0 \pm 0.00$ | $1000.0 \pm 0.00$ | $100.0 \pm 0.00$ |
| | NACP | $\mathbf{1.3 \pm 0.04}$ | $\mathbf{94.5 \pm 0.62}$ | $\mathbf{4.8 \pm 0.13}$ | $\mathbf{93.0 \pm 0.48}$ | $\mathbf{9.0 \pm 0.50}$ | $\mathbf{93.6 \pm 0.70}$ | $\mathbf{7.1 \pm 0.13}$ | $\mathbf{91.9 \pm 0.34}$ |
| | NACP (w/o Δ) | $1.1 \pm 0.02$ | $90.1 \pm 0.69$ | $4.0 \pm 0.11$ | $89.9 \pm 0.55$ | $6.7 \pm 0.27$ | $89.3 \pm 0.80$ | $6.3 \pm 0.10$ | $90.0 \pm 0.36$ |
| HPS | CP (Oracle) | $0.9 \pm 0.01$ | $90.0 \pm 0.59$ | $2.0 \pm 0.03$ | $90.0 \pm 0.43$ | $3.8 \pm 0.13$ | $90.02 \pm 0.58$ | $3.6 \pm 0.07$ | $90.0 \pm 0.28$ |
| | Noisy-CP | $5.1 \pm 0.18$ | $99.8 \pm 0.04$ | $50.1 \pm 1.34$ | $99.9 \pm 0.02$ | $98.3 \pm 3.80$ | $99.8 \pm 0.05$ | $501.3 \pm 10.2$ | $100.0 \pm 0.01$ |
| | NR-CP | $1.2 \pm 0.01$ | $97.8 \pm 0.12$ | $11.7 \pm 0.37$ | $98.9 \pm 0.08$ | $28.1 \pm 1.20$ | $98.1 \pm 0.15$ | $55.9 \pm 0.87$ | $99.1 \pm 0.02$ |
| | ACNL | $1.1 \pm 0.03$ | $96.0 \pm 0.59$ | $100.0 \pm 0.00$ | $100.0 \pm 0.00$ | $200.0 \pm 0.00$ | $100.0 \pm 0.00$ | $1000.0 \pm 0.00$ | $100.0 \pm 0.00$ |
| | NACP | $\mathbf{1.1 \pm 0.01}$ | $\mathbf{95.9 \pm 0.18}$ | $\mathbf{2.5 \pm 0.09}$ | $\mathbf{93.0 \pm 0.52}$ | $\mathbf{7.0 \pm 0.87}$ | $\mathbf{93.6 \pm 0.72}$ | $\mathbf{4.8 \pm 0.23}$ | $\mathbf{91.9 \pm 0.36}$ |
| | NACP (w/o Δ) | $0.9 \pm 0.02$ | $90.0 \pm 0.75$ | $2.0 \pm 0.06$ | $89.9 \pm 0.56$ | $3.5 \pm 0.24$ | $89.3 \pm 0.80$ | $3.6 \pm 0.14$ | $90.0 \pm 0.38$ |

Table 4: Finite sample correction terms $\Delta$ of NACP and ACNL (Sesia et al., 2023) for several datasets and two noise levels, $n$ is the size of the validation set.

| Dataset | $n$ | #classes | NACP | | ACNL | |
|---|---|---|---|---|---|---|
| | | | $\epsilon = 0.1$ | $\epsilon = 0.2$ | $\epsilon = 0.1$ | $\epsilon = 0.2$ |
| CIFAR-10 | 5000 | 10 | 0.0351 | **0.0431** | **0.0312** | 0.0594 |
| CIFAR-100 | 10000 | 100 | **0.0248** | **0.0305** | 0.0769 | 0.1634 |
| TinyImagenet | 5000 | 200 | **0.0351** | **0.0431** | 0.1745 | 0.3819 |
| ImageNet | 25000 | 1000 | **0.0157** | **0.0193** | 0.1936 | 0.4658 |

labels. The two evaluation metrics are formally defined as:

$$\text{size} = \frac{1}{n} \sum_i |C(x_i)|, \qquad \text{coverage} = \frac{1}{n} \sum_i \mathbf{1}(y_i \in C(x_i))$$

such that $n$ is the size of the test set. We report the mean and standard deviation over 1000 random splits.

**Datasets.** We present results on several publicly available medical and scenary imaging and classification datasets. **TissuMNIST** (Yang et al., 2021; 2023): This dataset contains 236,386 human kidney cortex cells, organized into 8 categories. Each gray-scale image is $32 \times 32 \times 7$ pixels. The 2D projections were obtained by taking the maximum pixel value along the axial-axis of each pixel, and were resized into $28 \times 28$ gray-scale images (Woloshuk et al., 2021). **PathMNIST** (Yang et al., 2023): This dataset contains 97,176 images of colon pathology with nine classes. The image size is $28 \times 28$. Here, we used a train/validation/test split of 89,996/3,590/3,590 images. **OrganSMNIST** (Yang et al., 2023): This dataset contains 25,221 images of abdominal CT in eleven classes. The images are $28 \times 28$ in size. Here, we used a train/validation/test split of 13,940/2,452/8,829 images. **OrganAMNIST** and **OrganCMNIST** are similar datasets, the differences of Organ{A,C,S}MNIST are the views and dataset size. We also show results on four standard scenery image datasets **CIFAR-10**, **CIFAR-100** (Krizhevsky et al., 2009), **Tiny-ImageNet**, and **ImageNet**.

**Implementation details.** Each task was trained by fine-tuning on a pre-trained ResNet-18 (He et al., 2016) network. The models were taken from the PyTorch site[4]. We selected this network architecture because of its widespread use in classification problems. The last fully connected layer output size was modified to fit the corresponding number of classes for each dataset. For PathMNIST, TissueMNIST, OrganSMNIST, OrganCMNIST, and OrganAMNIST we used publicly available checkpoints[5]. Also for the standard dataset evaluated in Table 3 we used publicly available checkpoints. For each dataset, we combined the validation and test sets and then constructed 1000 different splits where 50% was used for the calibration phase and 50% was used for testing. In all our experiments we used $\delta = 0.001$.

**CP results on medical datasets.** Table 1 reports the noisy label calibration results for randomized APS and RAPS. In all cases, we used $1-\alpha = 0.9$ and a noise level of $\epsilon = .2$. The results indicate that in the case of a validation set with noisy labels, the Noisy-CP threshold became larger to facilitate the uncertainty induced by the noisy labels. This yielded larger prediction sets and the coverage was higher than the target coverage which was set to $90\%$. The NACP

---

[4]https://pytorch.org/vision/stable/models.html
[5]https://github.com/MedMNIST/MedMNIST

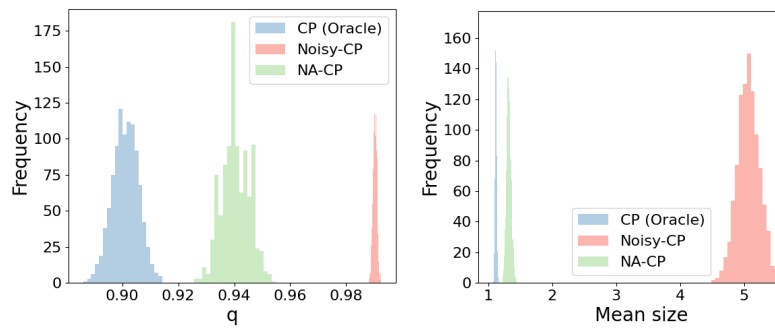

Figure 1: Histograms showing the distribution of (1) $q$ and (2) mean size values across 1000 splits on CIFAR-10 for the rand-APS score and the CP (Oracle), Noisy-CP, and NACP methods.

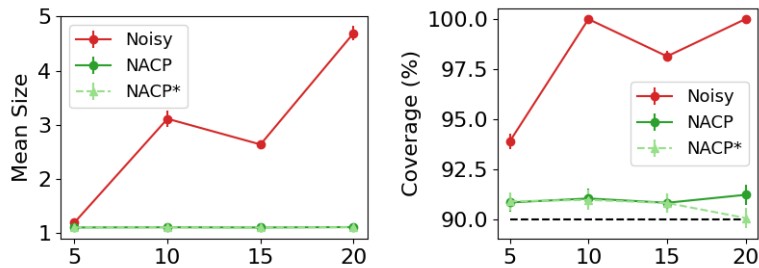

Figure 2: Mean size and coverage as a function of the noise level using rand-APS for PathMNIST and $1-\alpha = 0.9$. The calibration set size is $n = 8592$ (Table 2).

Table 5: Rand-APS calibration results for $1-\alpha = 0.9$ on CIFAR-100 dataset and two noise models. We report the mean and the std over 1000 different splits.

| CP Method | Neighborhood noise | | Random noise | |
|---|---|---|---|---|
| | size $\downarrow$ | coverage (%) | size $\downarrow$ | coverage (%) |
| CP (oracle) | $6.48 \pm 0.19$ | $90.01 \pm 0.41$ | $6.48 \pm 0.19$ | $90.01 \pm 0.41$ |
| Noisy-CP | $48.89 \pm 1.13$ | $99.80 \pm 0.04$ | $50.25 \pm 1.37$ | $99.82 \pm 0.04$ |
| NR-CP | $12.82 \pm 0.36$ | $95.62 \pm 0.21$ | $37.01 \pm 0.88$ | $99.53 \pm 0.06$ |
| NACP (w/o $\Delta$) | $\mathbf{6.52 \pm 0.22}$ | $90.03 \pm 0.47$ | $\mathbf{6.45 \pm 0.30}$ | $89.97 \pm 0.57$ |

method, which was aware of the noise rate, yielded better and almost on-par results with the noise-free scenario in terms of the prediction set size. Finally, NACP outperformed the NR-CP and the ACNL methods. We can also see that in practice, even without adding $\Delta$, the NACP (w/o $\Delta$) method obtained the required coverage. A comparison of the finite sample correction terms $\Delta$ obtained by NACP and ACNL is shown in Table 2.

**CP results on datasets with a large number of classes.** Following Theorem 3.3, we expect the gain in performance when using NACP versus Noisy-CP to increase with the number of classes. To validate this empirically, we tested our method on four standard publicly available datasets, CIFAR-10, CIFAR-100, Tiny-ImageNet and ImageNet. In all cases, we used $1-\alpha = 0.9$ and a noise level of $\epsilon = .2$. Table 3 details the results across 3 different conformal prediction scores, in all cases NACP achieved superior results. The prediction sets for Noisy-CP are usually half of the total classes, showing that Noisy-CP cannot handle this setting properly. Here for CIFAR-100, Tiny-ImageNet, and ImageNet the ACNL method (Sesia et al., 2023) completely failed due to the large number of classes and the relatively small number of samples per class. A comparison of the finite sample correction terms $\Delta$ obtained by NACP and ACNL is shown in Table 4. Note that if $1 - \alpha + \Delta > 1$, the prediction set should include all the classes and thus it becomes useless. We can see in Table 4 that this is the case for ACNL in datasets with a large number of classes.

**Different noise levels and End-to-end CP with noise level estimation.** Next, we address one of the basic limitations of our work, that the noise level needs to be known in advance. We show that our approach works well even when we use an estimation fo the noise level. To do so, we combined our NACP with a noise-robust network training that

estimated the noise level $\epsilon$ as part of the training (Li et al., 2021). Fig. 2 reports the mean size and coverage results across different noise levels $\epsilon$ and different conformal prediction methods on the PathMNIST dataset. We denote the NACP variant in which $\epsilon$ was estimated by NACP$^*$. The results show that as the noise rate increased, the noisy labels corrupted the calibration of the network more, as reflected in a larger mean size and higher coverage, whereas our method, even in the case where $\epsilon$ was estimated achieved much better calibration results which were on par with the result of the noise-free CP.

**General noise transition matrix.** Finally, we evaluate NACP on two common general noise matrices: Neighborhood noise and Random noise (see details in the supplementary). While existing final sample terms bounds are not effective, in practice NACP (without a finite sample correction) achieves the required coverage guarantee and the average prediction size is similar to the one obtained by the noise-free CP. We observe the same pattern when using uniform noise. This indicates that the current coverage guarantee bounds are too conservative. Table 5 shows the results on the CIFAR-100 dataset and the rand-APS technique when using NACP without finite sample correction term $\Delta$. Results show a clear dominance of NACP over Noisy-CP and NRCP on the two different noise models, presenting the robustness of NACP across various noise models. ACNL (without the finite sample term) achieves here similar results.

## 5 CONCLUSIONS

We presented a procedure that applies the Conformal Prediction algorithm on a validation set with noisy labels. We first presented our method in the simpler case of a uniform noise model and then extended it to a general noise matrix. We showed that if the noise level is given, we can find the noise-free calibration threshold without access to clean data by using the noisy-label data. We provided a finite-sample coverage guarantee for the case of uniform noise. We showed that in case of uniform noise our method outperforms current noisy CP methods by a large margin, in terms of the average size of the prediction set, while maintaining the required coverage. In all the experiments we conducted the finite sample term was effective and yielded a small prediction set. We showed, however, that even without adding the finite term we obtained the required coverage. This indicates that the current coverage guarantee analysis is too conservative and there is room for future research to improve it. In this study, we focused on noise models that assume the noisy label and the input image are independent, given the true label. In a more general noise model, the label corruption process also depends on the input features.

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

## A    APPENDIX / SUPPLEMENTAL MATERIAL

### A.1    GENERAL NOISE MATRICES

We define the two common general noise matrices. The Neighborhood noise as:

$$P_{i,j} = p(\tilde{y} = j | y = i) = \begin{cases} \xi & \text{if } i = j \\ 1 - \xi & \text{if } |i - j| = 1 \text{ and } i \in (1, k) \\ (1 - \xi)/2 & \text{if } |i - j| = 1 \text{ and } i \notin (1, k) \\ 0 & \text{otherwise} \end{cases} \tag{19}$$

The Random noise is defined as: first, on the diagonal, we have $\xi$. next, for each line (aka $\forall i$) the rest of the values (i.e. $k - 1$ items) are sampled from a random distribution $u_i$ vector of size $k - 1$ and then normalized to sum up to $1 - \xi$ to keep the matrix a probability matrix.

$$P_{i,j} = p(\tilde{y} = j | y = i) = \begin{cases} \xi & \text{if } i = j \\ (1 - \xi) \cdot \frac{u_i[j]}{\sum_{z \neq i} u_i[z]} & \text{otherwise} \end{cases} \tag{20}$$

In our experiments, we set $\xi$ such that the diagonal would be the same as the uniform noise experiments , i.e. $\xi = 1 - \epsilon + \epsilon/k$ where $\epsilon = 0, 2$.

### A.2    A FINITE SAMPLE TERM FOR THE CASE OF A GENERAL NOISE MATRIX

**Theorem A.1.** *Let $P$ be a general noise matrix. Given $\delta > 0$, define $\Delta = \|P^{-1}\|_\infty k \sqrt{\frac{\log(2k^2/\delta)}{2n}}$ where , $k$ is the number of classes and $n$ is the size of the noisy validation set. Then*

$$p(\sup_q |\hat{F}^c(q) - F^c(q)| > \Delta) < \delta.$$

*Proof.* From Eq. (18) we have $|\hat{F}^c(q) - F^c(q)| = |\operatorname{Tr}(P^{-1}\hat{M}_q) - \operatorname{Tr}(P^{-1}M_q)| = |\operatorname{Tr}(P^{-1}\Delta \hat{M}_q)|$ where $\Delta M_q = \hat{M}_q - M_q$. We first note that $M_q[i, j] = p(j \in C_q(x), \tilde{y} = i)$ is not a CDF but we can define one that agrees with it for $q \in (-\infty, C)$ which is the range of interest where $C$ is a constant that bound the score function $S(x, y)$ from above. We define $\tilde{S}_{ij}(x, \tilde{y}) = \begin{cases} S(x, j), & \text{if } \tilde{y} = i \\ C, & \text{if } \tilde{y} \neq i \end{cases}$ so $M_q[i, j] = p(\tilde{S}_{ij}(x, \tilde{y}) \leq q)$ for $q \in (-\infty, C)$. Now from the DKW theorem, we know that if we estimate a CDF using $n$ samples then with probability at least $1 - \delta$ we get a uniform bound on the error of size $\sqrt{\frac{\log(2/\delta)}{2n}}$. As we are estimating $k^2$ matrix elements we can use the union bound to get that with probability $1 - \delta$ the $\forall i, j, q \in (-\infty, C) : |\Delta M_q| \leq \sqrt{\frac{\log(2k^2/\delta)}{2n}}$. Now if we look at the infinity norm of $P^{-1}$, then $|(P^{-1}\Delta M_q)_{i,j}| \leq \|P^{-1}\|_\infty \sqrt{\frac{\log(2k^2/\delta)}{2n}}$. As the trace is the sum of $k$ such matrix entries, the total bound is $\kappa_\infty k \sqrt{\frac{\log(2k^2/\delta)}{2n}}$ for $q \in (-\infty, C)$. Since we know $F^c(q) = 1$ for $q \geq C$, we can set $\hat{F}^c(q) = 1$ for $q \geq C$ and get a bound for all $q \in \mathbb{R}$. □

### A.3    TABLE 1 WITH MODELS TRAINED WITH NOISY LABELS

Reproducing Table 1 by replacing models publicly available with models trained with noisy labels. We combined our NACP with a noise-robust network training that estimated the noise level $\epsilon$ as part of the training (Li et al., 2021). As a by-product of the noisy labels training procedure, we get a noise estimation that can be used in the calibration step. Therefore the following table has three additional rows denoted by $*$ corresponding to ACNL, NACP, and NACP (w/o $\Delta$) using $\tilde{\epsilon}$ instead of the true $\epsilon = 0.2$. The estimated noise $\tilde{\epsilon}$ for each model is detailed in Table 6 and the results are shown in Table 7.

### A.4    IMAGENET WITH DIFFERENT CALIBRATION SET SIZE

In the following experiment, we test the performance of various conformal prediction methods under noisy labels as a function of the calibration set size on the ImageNet dataset. Figure 3 shows the mean size and coverage as a function of

| Dataset | TissueMNIST | OrganSMNIST | OrganCMNIST | OrganAMNIST | PathMNIST |
|---|---|---|---|---|---|
| Estimated noise $\tilde{\epsilon}$ | 0.175 | 0.19 | 0.18 | 0.165 | 0.16 |

Table 6: Estimated noise using for the five medical datasets, given true noise $\epsilon = 0.2$.

Table 7: APS and RAPS (randomized versions) calibration results for $1-\alpha = 0.9$ and $\epsilon = 0.2$. We report the mean and the std over 1000 different splits. Bold for best result with theoretical guarantees.

| Dataset | CP Method | rand-APS | | rand-RAPS | |
|---|---|---|---|---|---|
| | | size ↓ | coverage (%) | size ↓ | coverage (%) |
| TissueMNIST | CP (Oracle) | $3.39 \pm 0.02$ | $90.02 \pm 0.21$ | $3.39 \pm 0.02$ | $90.02 \pm 0.20$ |
| | Noisy-CP | $5.03 \pm 0.03$ | $96.83 \pm 0.11$ | $5.04 \pm 0.03$ | $96.81 \pm 0.12$ |
| | NR-CP | $4.07 \pm 0.02$ | $93.74 \pm 0.12$ | $4.18 \pm 0.02$ | $94.21 \pm 0.12$ |
| | ACNL | $4.29 \pm 0.04$ | $94.63 \pm 0.20$ | $5.45 \pm 0.02$ | $97.61 \pm 0.08$ |
| | NACP | $\mathbf{3.65 \pm 0.03}$ | $\mathbf{91.67 \pm 0.25}$ | $\mathbf{3.65 \pm 0.03}$ | $\mathbf{91.66 \pm 0.25}$ |
| | NACP (w/o $\Delta$) | $3.39 \pm 0.03$ | $90.03 \pm 0.25$ | $3.39 \pm 0.03$ | $90.03 \pm 0.25$ |
| estimated $\epsilon$ | ACNL* | $4.15 \pm 0.04$ | $94.08 \pm 0.21$ | $4.18 \pm 0.02$ | $94.68 \pm 0.18$ |
| | NACP* | $\mathbf{3.64 \pm 0.03}$ | $\mathbf{91.60 \pm 0.24}$ | $\mathbf{3.64 \pm 0.03}$ | $\mathbf{91.61 \pm 0.24}$ |
| | NACP* (w/o $\Delta$) | $3.40 \pm 0.02$ | $90.06 \pm 0.23$ | $3.40 \pm 0.03$ | $90.05 \pm 0.25$ |
| OrganSMNIST | CP (Oracle) | $1.64 \pm 0.02$ | $90.04 \pm 0.58$ | $1.63 \pm 0.02$ | $90.02 \pm 0.60$ |
| | Noisy-CP | $5.59 \pm 0.18$ | $99.73 \pm 0.06$ | $5.58 \pm 0.19$ | $99.74 \pm 0.05$ |
| | NR-CP | $3.05 \pm 0.06$ | $98.23 \pm 0.12$ | $3.79 \pm 0.12$ | $99.21 \pm 0.12$ |
| | ACNL | $2.83 \pm 0.23$ | $97.77 \pm 0.48$ | $3.20 \pm 0.03$ | $97.90 \pm 0.03$ |
| | NACP | $\mathbf{1.95 \pm 0.06}$ | $\mathbf{93.93 \pm 0.64}$ | $\mathbf{1.94 \pm 0.06}$ | $\mathbf{93.97 \pm 0.60}$ |
| | NACP (w/o $\Delta$) | $1.63 \pm 0.03$ | $89.94 \pm 0.67$ | $1.62 \pm 0.03$ | $90.00 \pm 0.45$ |
| estimated $\epsilon$ | ACNL* | $2.63 \pm 0.17$ | $97.30 \pm 0.53$ | $3.08 \pm 0.04$ | $97.62 \pm 0.05$ |
| | NACP* | $\mathbf{1.94 \pm 0.05}$ | $\mathbf{93.88 \pm 0.54}$ | $\mathbf{1.93 \pm 0.05}$ | $\mathbf{93.87 \pm 0.58}$ |
| | NACP* (w/o $\Delta$) | $1.63 \pm 0.03$ | $89.86 \pm 0.69$ | $1.62 \pm 0.03$ | $89.86 \pm 0.59$ |
| OrganCMNIST | CP (Oracle) | $1.22 \pm 0.02$ | $90.08 \pm 0.52$ | $1.22 \pm 0.01$ | $90.01 \pm 0.66$ |
| | Noisy-CP | $5.50 \pm 0.20$ | $99.85 \pm 0.05$ | $5.58 \pm 0.19$ | $99.69 \pm 0.05$ |
| | NR-CP | $2.15 \pm 0.05$ | $98.5 \pm 0.16$ | $3.42 \pm 0.31$ | $99.57 \pm 0.08$ |
| | ACNL | $1.84 \pm 0.12$ | $97.49 \pm 0.55$ | $1.76 \pm 0.10$ | $97.41 \pm 0.50$ |
| | NACP | $\mathbf{1.40 \pm 0.03}$ | $\mathbf{94.00 \pm 0.59}$ | $\mathbf{1.40 \pm 0.03}$ | $\mathbf{93.97 \pm 0.63}$ |
| | NACP (w/o $\Delta$) | $1.21 \pm 0.02$ | $90.03 \pm 0.62$ | $1.24 \pm 0.02$ | $90.11 \pm 0.65$ |
| estimated $\epsilon$ | ACNL* | $1.66 \pm 0.08$ | $96.50 \pm 0.59$ | $1.51 \pm 0.05$ | $95.80 \pm 0.51$ |
| | NACP* | $\mathbf{1.39 \pm 0.03}$ | $\mathbf{93.70 \pm 0.61}$ | $\mathbf{1.38 \pm 0.03}$ | $\mathbf{93.61 \pm 0.54}$ |
| | NACP* (w/o $\Delta$) | $1.21 \pm 0.02$ | $89.71 \pm 0.65$ | $1.21 \pm 0.02$ | $89.70 \pm 0.76$ |
| OrganAMNIST | CP (Oracle) | $1.11 \pm 0.01$ | $89.98 \pm 0.38$ | $1.11 \pm 0.01$ | $90.00 \pm 0.36$ |
| | Noisy-CP | $5.48 \pm 0.12$ | $99.87 \pm 0.03$ | $5.52 \pm 0.11$ | $99.85 \pm 0.03$ |
| | NR-CP | $1.94 \pm 0.03$ | $98.59 \pm 0.08$ | $3.49 \pm 0.19$ | $99.65 \pm 0.06$ |
| | ACNL | $1.35 \pm 0.03$ | $95.54 \pm 0.42$ | $1.56 \pm 0.04$ | $96.54 \pm 0.39$ |
| | NACP | $\mathbf{1.19 \pm 0.01}$ | $\mathbf{92.53 \pm 0.44}$ | $\mathbf{1.19 \pm 0.01}$ | $\mathbf{92.51 \pm 0.37}$ |
| | NACP (w/o $\Delta$) | $1.11 \pm 0.01$ | $89.75 \pm 0.48$ | $1.11 \pm 0.01$ | $89.74 \pm 0.43$ |
| estimated $\epsilon$ | ACNL* | $1.28 \pm 0.02$ | $94.39 \pm 0.45$ | $1.32 \pm 0.03$ | $94.52 \pm 0.40$ |
| | NACP* | $\mathbf{1.19 \pm 0.01}$ | $\mathbf{92.33 \pm 0.35}$ | $\mathbf{1.18 \pm 0.01}$ | $\mathbf{92.27 \pm 0.40}$ |
| | NACP* (w/o $\Delta$) | $1.11 \pm 0.01$ | $89.78 \pm 0.39$ | $1.11 \pm 0.01$ | $89.74 \pm 0.42$ |
| PathMNIST | CP (Oracle) | $1.10 \pm 0.02$ | $90.03 \pm 0.47$ | $1.03 \pm 0.01$ | $90.00 \pm 0.45$ |
| | Noisy-CP | $4.81 \pm 0.15$ | $99.98 \pm 0.01$ | $4.89 \pm 0.18$ | $99.95 \pm 0.03$ |
| | NR-CP | $1.46 \pm 0.03$ | $98.33 \pm 0.19$ | $1.46 \pm 0.05$ | $99.20 \pm 0.11$ |
| | ACNL | $1.25 \pm 0.05$ | $95.08 \pm 0.40$ | $1.32 \pm 0.06$ | $95.30 \pm 0.51$ |
| | NACP | $\mathbf{1.14 \pm 0.03}$ | $\mathbf{93.42 \pm 0.40}$ | $\mathbf{1.12 \pm 0.01}$ | $\mathbf{93.48 \pm 0.50}$ |
| | NACP (w/o $\Delta$) | $1.10 \pm 0.02$ | $90.08 \pm 0.30$ | $1.04 \pm 0.02$ | $90.14 \pm 0.53$ |
| estimated $\epsilon$ | ACNL* | $1.19 \pm 0.02$ | $93.81 \pm 0.47$ | $1.23 \pm 0.03$ | $94.00 \pm 0.31$ |
| | NACP* | $\mathbf{1.17 \pm 0.02}$ | $\mathbf{93.08 \pm 0.47}$ | $\mathbf{1.10 \pm 0.01}$ | $\mathbf{93.10 \pm 0.45}$ |
| | NACP* (w/o $\Delta$) | $1.10 \pm 0.02$ | $90.09 \pm 0.50$ | $1.03 \pm 0.01$ | $90.07 \pm 0.49$ |

the calibration set size. In addition, the correction term $\Delta$ is depicted for ImageNet for each calibration set size. Results show that even with as little as 2500 images that correspond to 2.5 images per class the calibration results are almost on par with the oracle calibration given clean labels.

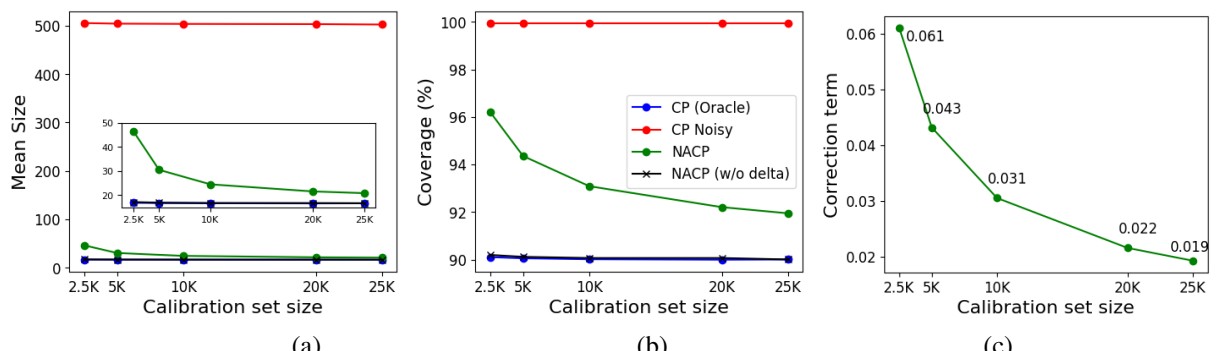

Figure 3: Noisy labels conformal prediction on ImageNet with different calibration set sizes. (a) Mean size (b) Coverage (%), and (c) Correction terms $\Delta$ as a function of calibration set size.

## A.5 NACP AGNOSTIC TO DIFFERENT MODEL ARCHITECTURES

Conformal prediction in general and our method NACP specifically has no assumption and is agnostic to the underlying model architecture. In the following section, we verify that by experimenting with ImageNet across different model architectures. Table 8 presents the results of applying conformal prediction with and without noisy labels on ResNet18, ResNet50, DenseNet121, ViT-B16 (Vision transformer).

Table 8: CP calibration results on ImageNet and various model architectures for $1-\alpha = 0.9$ and $\epsilon = 0.2$. We report the mean and the std over 1000 different splits. Bold for best result with theoretical guarantees.

| Dataset | CP Method | ResNet-18 size ↓ | ResNet-18 coverage(%) | ResNet-50 size ↓ | ResNet-50 coverage(%) | DenseNet121 size ↓ | DenseNet121 coverage(%) | ViT-B16 size ↓ | ViT-B16 coverage(%) |
|---|---|---|---|---|---|---|---|---|---|
| rand-APS | CP (Oracle) | $16.6 \pm 0.33$ | $90.0 \pm 0.26$ | $13.9 \pm 0.34$ | $90.0 \pm 0.28$ | $12.0 \pm 0.28$ | $90.0 \pm 0.27$ | $10.7 \pm 0.38$ | $90.0 \pm 0.25$ |
| | Noisy-CP | $502.6 \pm 8.56$ | $99.9 \pm 0.01$ | $505.5 \pm 8.11$ | $99.9 \pm 0.01$ | $502.8 \pm 8.46$ | $99.9 \pm 0.01$ | $506.8 \pm 8.14$ | $99.8 \pm 0.02$ |
| | ACNL | $1000.0 \pm 0.00$ | $100.0 \pm 0.00$ | $1000.0 \pm 0.00$ | $100.0 \pm 0.00$ | $1000.0 \pm 0.00$ | $100.0 \pm 0.00$ | $1000.0 \pm 0.00$ | $100.0 \pm 0.00$ |
| | NACP | $\mathbf{20.9 \pm 0.72}$ | $\mathbf{91.9 \pm 0.32}$ | $\mathbf{17.4 \pm 0.62}$ | $\mathbf{91.9 \pm 0.36}$ | $\mathbf{15.1 \pm 0.55}$ | $\mathbf{91.9 \pm 0.34}$ | $\mathbf{15.5 \pm 0.81}$ | $\mathbf{91.9 \pm 0.31}$ |
| | NACP (w/o $\Delta$) | $16.7 \pm 0.51$ | $90.0 \pm 0.34$ | $13.9 \pm 0.47$ | $90.0 \pm 0.37$ | $12.0 \pm 0.38$ | $90.0 \pm 0.34$ | $10.7 \pm 0.55$ | $90.0 \pm 0.35$ |
| rand-RAPS | CP (Oracle) | $6.3 \pm 0.06$ | $90.0 \pm 0.27$ | $4.5 \pm 0.05$ | $89.9 \pm 0.29$ | $4.7 \pm 0.06$ | $90.0 \pm 0.26$ | $2.6 \pm 0.04$ | $90.0 \pm 0.25$ |
| | Noisy-CP | $501.6 \pm 8.51$ | $99.9 \pm 0.01$ | $501.1 \pm 8.85$ | $99.9 \pm 0.01$ | $501.9 \pm 8.80$ | $99.9 \pm 0.01$ | $505.8 \pm 7.90$ | $99.9 \pm 0.01$ |
| | ACNL | $1000.0 \pm 0.00$ | $100.0 \pm 0.00$ | $1000.0 \pm 0.00$ | $100.0 \pm 0.00$ | $1000.0 \pm 0.00$ | $100.0 \pm 0.00$ | $1000.0 \pm 0.00$ | $100.0 \pm 0.00$ |
| | NACP | $\mathbf{7.1 \pm 0.13}$ | $\mathbf{91.9 \pm 0.34}$ | $\mathbf{5.0 \pm 0.08}$ | $\mathbf{91.9 \pm 0.34}$ | $\mathbf{5.3 \pm 0.10}$ | $\mathbf{92.0 \pm 0.35}$ | $\mathbf{2.9 \pm 0.07}$ | $\mathbf{92.0 \pm 0.30}$ |
| | NACP (w/o $\Delta$) | $6.3 \pm 0.10$ | $90.0 \pm 0.36$ | $4.5 \pm 0.06$ | $90.0 \pm 0.35$ | $4.7 \pm 0.08$ | $90.0 \pm 0.36$ | $2.6 \pm 0.05$ | $90.0 \pm 0.36$ |
| HPS | CP (Oracle) | $3.6 \pm 0.07$ | $90.0 \pm 0.28$ | $2.0 \pm 0.03$ | $90.0 \pm 0.28$ | $2.4 \pm 0.03$ | $90.0 \pm 0.25$ | $1.5 \pm 0.02$ | $90.0 \pm 0.26$ |
| | Noisy-CP | $501.3 \pm 10.2$ | $100.0 \pm 0.01$ | $502.4 \pm 9.50$ | $99.9 \pm 0.01$ | $502.3 \pm 10.3$ | $99.9 \pm 0.20$ | $504.3 \pm 8.19$ | $99.9 \pm 0.01$ |
| | ACNL | $1000.0 \pm 0.00$ | $100.0 \pm 0.00$ | $1000.0 \pm 0.00$ | $100.0 \pm 0.00$ | $1000.0 \pm 0.00$ | $100.0 \pm 0.00$ | $1000.0 \pm 0.00$ | $100.0 \pm 0.00$ |
| | NACP | $\mathbf{4.8 \pm 0.23}$ | $\mathbf{91.9 \pm 0.36}$ | $\mathbf{2.6 \pm 0.10}$ | $\mathbf{91.9 \pm 0.37}$ | $\mathbf{3.1 \pm 0.12}$ | $\mathbf{91.9 \pm 0.34}$ | $\mathbf{1.7 \pm 0.04}$ | $\mathbf{91.9 \pm 0.33}$ |
| | NACP (w/o $\Delta$) | $3.6 \pm 0.14$ | $90.0 \pm 0.38$ | $2.1 \pm 0.06$ | $90.0 \pm 0.38$ | $2.4 \pm 0.07$ | $90.0 \pm 0.34$ | $1.5 \pm 0.03$ | $90.0 \pm 0.35$ |

