# OpenReview forum: "Estimating the conformal prediction threshold from noisy labels"
_ICLR.cc/2025/Conference — Submitted to ICLR 2025_

### Official Review · Reviewer_81YJ · 2024-11-03

**Soundness:** 3
**Presentation:** 3
**Contribution:** 3
**Rating:** 6
**Confidence:** 3

**Summary:**

This paper proposes a new method called Noise-Aware Conformal Prediction (NACP) to address the calibration issue when only noisy-labeled validation data is available. The method can estimate noise-free conformal thresholds from noisy data without requiring information about the underlying data distribution or classifier mechanisms. Experiments on medical and natural image datasets demonstrate that NACP significantly outperforms existing methods, especially for large-scale classification tasks.

**Strengths:**

- The paper is well-written. The authors provide theoretical coverage guarantees for their approach, particularly for uniform noise cases, and extend the framework to general noise transition matrices.
- The extensive experiments demonstrate superior performance of NACP on various classification tasks, where prior methods fail to maintain reasonable prediction set sizes.

**Weaknesses:**

- The work lacks analysis on calibration set sizes, particularly for large-scale datasets like ImageNet. It remains unclear how the method performs with different validation set sizes and what is the minimum required samples per class for reliable performance.
- The experiments were conducted only with ResNet-18 architecture, leaving it unclear whether the method's effectiveness generalizes across different network architectures such as deeper ResNets or Vision Transformers.

**Questions:**

Could you further explain how the noise transition matrices would be obtained in practice? While the paper shows promising results with different noise matrices, it would be helpful to clarify whether additional data is needed for estimation and how estimation errors might affect performance.

---

> ### Author Response · Authors · 2024-11-20
> **Reply to reviewer 81Yj**
>
> W1: We added an analysis of ImageNet's calibration set sizes (see Fig 3 in section A.4). As expected, performance is monotonically improved as a function of the validation set size.
>
> w2: We added experiments with other architectures such as deeper ResNets and Vision Transformers, and observed the same trends (see Table 8 in section A.5).
>
> Q: Could you further explain how the noise transition matrices would be obtained in practice?
>
> A: When training a model with noisy-labeled data, if the noise matrix is unknown a popular noise-robust approach is to jointly learn the noise matrix and the model parameters (see e.q. Li et al, 2021).  In Section A.3 we show the results of jointly learning the model and the noise matrix with noisy data and then using the learned noise model in our NACP calibration approach.

---

### Official Review · Reviewer_4Nej · 2024-11-03

**Soundness:** 3
**Presentation:** 3
**Contribution:** 3
**Rating:** 8
**Confidence:** 3

**Summary:**

This paper addresses the problem of conformal prediction when the validation set has noisy labels. Algorithms are proposed for two different types of label noises: a uniform noise level $\epsilon$ or a more general noise matrix. In both cases, the noise condition (level/matrix) are assumed to be known. The algorithms will depend on such noise condition.  The paper analyzes the coverage and sample size required from the validation set, and shows the superiority of the proposed method compared with existing ones, e.g., (Sesia et al. 2023), especially regarding data with large label set and small sizes per class.

Overall, the paper is well written with well justified algorithms and strong empirical results. However, I do have the following concerns/questions that need to be clarified.

1, I have some concerns over the problem setup. If I am not mistaken, the algorithms assume a good model, i.e., the model is trained on clean labels. But if the validation set is already noisy, it is hard to assume the training data is clean, right? So ideally, the method should be discussing a conformal prediction training and validating on noisy labels. Even if this is too much to ask, there should be experimental evaluation on how the proposed algorithm (and the baselines) perform on the validation set when the model is trained on equally noisy training set.

2, The proposed methods assume a known noise condition. This is unrealistic. The paper justifies by saying that one could estimate the noise level or the noise matrix through training set. But this would assume a noisy training set in the first place. If the noise condition is estimated through the validation set, there is a question of how reliable the noise condition estimation. In particular, the smaller the sample size is per class, the less reliable these noise condition estimation will be.

3, the above concern should be discussed regarding other baselines. For example, does SoTA methods (e.g., Sesia et al. 2023) depend on the oracle noise condition? In experiments, if one has to estimate noise using the noisy validation set, how will the proposed method compare with Sesia et al. 2023, and compare with a vanilla conformal prediction?

4, the statement that the second algorithm can handle general noise is an over-statement. Even with the noise matrix, the paper is still assuming the noise is iid. There are plenty of research on label noise problem assuming the noise is not iid, e.g., feature dependent. I understand this is still early stage of the problem, so it is OK not tackling such noise models. But it would be important to mention this in the paper.

Minor errors:
Line 88 – C(x) --> C_q(x)
Equation 16, minor typo?

**Strengths:**

Solid algorithm and analysis.

Strong empirical results.

**Weaknesses:**

Problem settings and assumptions.

Potential unfair comparisons with existing methods.

See Summary for details.

**Questions:**

See Summary for details.

---

> ### Author Response · Authors · 2024-11-20
> **Reply to Reviewer  4Nej**
>
> Q1+Q2: The conformal calibration step is agnostic to the underlying trained model, which is why we decoupled the calibration experiments from the training process, treating the model as a black box. To address the reviewers’ concerns, we have incorporated more realistic experiments in Section A.3. These experiments demonstrate the end-to-end process of jointly training the model and estimating the noise parameter $\epsilon$ directly on noisy data. Subsequently, the learned noise model is integrated into the calibration step. The results validate that the effectiveness of NACP persists also under these conditions.
>
> Q3: All other methods, including those proposed by Sesia et al. (2023), depend on the oracle noise condition or assume access to clean data. The experiment in Section A.3 (Table 7) compares NACP to other methods under scenarios where the noise is either known or estimated.
>
> Q4: We added a comment in the text to clarify this point.

---

> > ### Comment · Reviewer_4Nej · 2024-11-27
> > **Increasing my score**
> >
> > The rebuttal with new experiments addressed my concern. The setting of learning robust-to-noise model while estimating noise parameter is reasonably close to reality. There is still concerns as to whether only dealing with the simple noise model ($\epsilon$ instead of a noise transition matrix) is sufficient, and I am also worried whether the theory will still hold if the model is trained on noisy data. But I think it is OK as this paper is one of the few early attempts on this direction.
> >
> > Overall, I think conformal prediction is a good task to tackle for noisy label setting, and this paper has made good progress in both theory and practice. I therefore raise my score to 8.

---

### Official Review · Reviewer_KQBp · 2024-11-04

**Soundness:** 2
**Presentation:** 2
**Contribution:** 2
**Rating:** 5
**Confidence:** 4

**Summary:**

This paper studies the behavior of conformal prediction in the case of validation data with noisy labels and develop a method for estimating the noise-free CP threshold in the context of noisy validation data. The noise-aware conformal prediction achieves better performance in average set size than other state-of-the-art algorithms.

**Strengths:**

Extensive experiments for performance comparison to show the advantage of NACP in reducing the average set size while strictly maintaining the validity of coverage;
Consideration of several noise models, including random noise and noise matrix, and this greatly expands the potential application of the proposed method.
Establishement of an end-to-end CP for estimating noise level in the training of neural network.

**Weaknesses:**

If you can demonstrate NACP in a more practical setting rather than a simulated environment, this will make the implication of the theorectical contribution of this paper more convincing.
What is the connection between the two noise models (uniform noise distribution, noise matrix) and practical learning tasks? How should I choose which noise model I should use in practical problems, by trial and error or other approaches?
in Fig. 2, can you also include the set size assocaited with each conformal predictors as this is also an important aspect to evaluate CP performance.

**Questions:**

See above.

---

> ### Author Response · Authors · 2024-11-20
> **reply to reviewer KQBp**
>
> The uniform noise model we used is a standard noise modeling. It is indeed very simple (but still relevant in some cases, such as differential privacy). However, even in this simple case, the problem is still not completely solved and we managed to improve SOTA results.
>
> We added more realistic experiments in Section A.3 where we show the results of learning the model and the noise parameter epsilon with noisy data and then using the learned noise model in the calibration step.
>
> In Fig. 2, We added the set size associated with the conformal predictor.

---

### Official Review · Reviewer_9NiR · 2024-11-04

**Soundness:** 3
**Presentation:** 3
**Contribution:** 1
**Rating:** 3
**Confidence:** 3

**Summary:**

This paper proposes a new method for calibrating conformal predictions when using a validation set with noisy labels. While conformity scores are based on previous research, the authors introduce a novel approach to calibrate these scores in the presence of noisy-labeled data in the validation set. This paper mainly addresses the cases where the noise distribution and its parameters are known. The main idea is to utilize the known noise distribution to reconstruct the clean distribution of the conformity scores.

**Strengths:**

- The proposed calibration method is applicable to a wide range of conformity scores.
- The authors provide a proof that the proposed calibration method meets coverage guarantees.
- Experimental results show that the prediction sets produced by the proposed method are significantly smaller than those generated by other existing methods on the given dataset and in the specified conditions.

**Weaknesses:**

- The main assumptions are highly restrictive. The authors assume that both the noise distribution and its parameters are known, and that the training data is either clean or uses pre-trained checkpoints without noisy data, while only the validation data contains noise. This scenario seems uncommon, which may limit the practical applicability of the proposed method.
- Additional real-world experimental studies could strengthen the motivation behind the proposed method. For example, clarifying real-world scenarios where the validation dataset is noisy or intentionally corrupted with the known noise distribution would be beneficial. Although the authors conducted experiments on popular classification datasets, the approach remains somewhat unconvincing, since these experiments could be considered as simulations with artificially injected noise. While the method demonstrates superior performance on these datasets, it would be more compelling if there were real-world examples in which the validation data is noisy with a known distribution and parameters. Without such context, the motivation for this method remains somewhat unclear.

**Questions:**

- While the meaning of $|C_q|$ and $C_q$ can be inferred from the context, a clear definition of these notations would improve readability.
- Although $\widehat{F}^c$ becomes more complex in the noisy case, it still depends only on the indicator function and the count of elements in $C_q$. It also appears to be piecewise constant, with potentially multiple values of $q$ satisfying $\widehat{F}^c(q)=1-\alpha$. In such cases how do you select $q$?
- If the authors intended that the clean label is corrupted into another label with probability of $\varepsilon$, equation (4) might need correction. Currently, equation (4) leads to a noisy label with probability of $\frac{k-1}{k}\varepsilon$ and clean label with probability of $1-\frac{k-1}{k}\varepsilon$.

---

> ### Author Response · Authors · 2024-11-20
> **Reply to reviewer 9NIR**
>
> First, we note that we do not assume clean training data, we simply used a classifier trained on clean data for convenience as our focus was on the post-training calibration. We added an experiment that reproduces Table 1, this time using models trained with noisy labels to better reflect real-world scenarios. Furthermore, we discussed how the noise matrix can be estimated during the training phase. We added additional results with estimated noisy labels in Section A.3, Table 7.
>
> Q1; in line 88 we  changed $C(x)$ to $C_q(x) $
>
> Q2: F(q) is almost a monotone function of q. In most cases, there was a unique q and when there were two solutions they were very close to each other (difference less than 0.001), and selecting one of the q's had no effect on the results.
>
> Q3: In the standard formulation of uniform noise model epsilon is the probability of replacing the correct label with a uniformly sampled label. The probability of label error is indeed (k-1)$\epsilon$\k. In a general noise matrix model, the probability that a label remains correct depends also on the class frequencies (which in the case of label noise cannot be directly inferred from the noisy data). One of the strengths of our method is that we don't need to know the class frequencies (while the ACNL method needs to know or estimate them).

---

> > ### Comment · Reviewer_9NiR · 2024-11-27
> >
> > Dear Authors,
> >
> > Thank you for addressing my major concern by incorporating additional experiments and discussing the estimation of noise parameters.
> >
> > While the additional experiments demonstrate the proposed method's empirical applicability to alternative scenarios, the most of the study still relies on unrealistic assumptions, particularly regarding theoretical guarantees. Although the results of the additional experiments are promising, experimental evidence alone is insufficient to show that the proposed method is theoretically sound without the known noise.
> >
> > Therefore, I will retain my current rating.

---

> > > ### Author Response · Authors · 2024-11-28
> > > **realistic noise model assumptions**
> > >
> > > Thank you for your reply.
> > >
> > > (a)  A known noise model is a realistic setting for problems such as differential privacy where the noise is artificially added. We are the first to establish an effective coverage guarantee for this case.
> > >
> > > (b) In case the noise model is unknown, while the coverage guarantee term may be off due to the error in estimation, we still show empirically how our method does handle this challenging realistic scenario. Thus, it is still of use for practitioners as no current CP method exists now that can give an effective coverage guarantee even when the noise model is known. As such this work is still a major step in solving CP with noisy labels and we believe it shouldn't be rejected simply for not solving the issue completely.

---

> > > > ### Comment · Reviewer_9NiR · 2024-12-03
> > > >
> > > > Thank you for your detailed rebuttal and for addressing the reviewers' concerns with additional experiments and clarifications. I acknowledge the effort you have made in advancing the study of conformal prediction under noisy label conditions.
> > > >
> > > > However, as raised by several reviewers, the primary assumption of a known noise distribution and its parameters remains a critical limitation that restricts the broader acceptance and applicability of this work. Considering that most noisy label problems originate from real-world situations where labels are corrupted for unknown reasons, practitioners aiming to quantify the uncertainty of their models often face challenges stemming from unknown noise. Although the authors suggest that the assumption of a known noise model is realistic in contexts such as differential privacy, the submission does not include detailed experiments within this domain to substantiate the claim.
> > > >
> > > > To address these limitations, the authors added promising experiments that demonstrate the potential for the proposed approach under unknown noise assumptions. However, experimental potential alone is not sufficient for acceptance in the context of uncertainty quantification. The theoretical coverage guarantee for the unknown noise problem is essential (including the uncertainty of $\epsilon$, which is the most important part).
> > > >
> > > > Regardless of this paper's acceptance outcome, I strongly encourage the authors to continue their exploration of conformal prediction under noisy label conditions with more realistic assumptions. By addressing the unknown noise problem more theoretically, I believe this work could make a significant contribution to this field.
> > > >
> > > > Therefore, I have ultimately decided to maintain my initial score.

---

### Official Review · Reviewer_VNWi · 2024-11-04

**Soundness:** 2
**Presentation:** 3
**Contribution:** 2
**Rating:** 5
**Confidence:** 3

**Summary:**

This paper introduces a new conformal prediction framework for classification that is able to handle label noise. Specifically, given known uniform label noise, the authors present an algorithm that can achieve the desired coverage for the underlying unobserved clean data. They also extend their approach to handle general noise using a known noise matrix. Through various experiments, they demonstrate that their approach outperforms existing methods for managing label noise in CP.

**Strengths:**

The paper is well-structured and easy to follow and understand, addressing the intriguing problem of conformal prediction sets in the presence of noisy data.

**Weaknesses:**

1. My primary concern relates to the noise assumption. Firstly, it's unclear how realistic the assumption of uniform label noise is in practical applications. Additionally, assuming that the noise parameter is known or can be accurately estimated in real-world scenarios doesn't seem entirely realistic (for both uniform and general noise).

2. In the experiments, different values of the noise parameter weren't tested to show how the model performs under varying noise levels (e.g., extending beyond $\epsilon > 0.2$).

3. For the results of the general noise model presented in Table 5, it would be fairer—particularly for the NR-CP approach—to include NACP with $\Delta$ as well, as these two methods are specifically aimed at recovering finite-sample coverage guarantees under noise.

**Questions:**

1. Does the grid search for finding q in [q_1, q_2] yield a unique result? If yes, why? If not, what might this imply?


2. According to the experimental results, it appears that NACP without $\Delta$ performs best—achieving the required coverage while maintaining the smallest average set size in almost all experiments. Could the authors elaborate on why this is the case?

3. While it’s valuable to see where the proposed method is effective, in what situations does it fail?

---

> ### Author Response · Authors · 2024-11-20
> **reply to reviewer VNWi**
>
> W1) Noise assumption. We added an experiment with a more realistic setup. We reproduced Table 1 with models that were trained with noisy labels, and the noise was estimated from the data. The results were added to Section A.3, Table 7. Regarding uniform noise, we agree that it is unrealistic [except for cases like differential privacy where noise is artificially injected], which is why we expand to the general noise model. However, even in this simple setting the CP calibration problem is not completely solved and we managed to improve SOTA results.
>
> W2) Noise level. We chose a working point of $\epsilon$=0.2 because on one hand it is a high noise level where ignoring the noise yields meaningless CP calibration results and on the other hand it is a noise level that can still occur in real-life situations.  We do address other noise levels in Tables 2 and 4 and Figure 2.
>
> W3) Note that NR-CP does not provide any coverage guarantees nor includes a correction term, potentially giving NR-CP an advantage in other tables. Hence comparing its performance to NACP, w/o finite sample term is a fair comparison.
>
> Q1) Grid search. F(q) is almost a monotone function of q. In most cases, there was a unique q and when there were two solutions we took the larger.  The solutions were very close to each other (difference less than 0.001), and selecting one of the q's had no effect on the results.
>
> Q2) NACP without delta performs best. The finite sample theoretical coverage guarantee provided by NACP was computed using the general DKW inequality which is a worst-case bound. In our example, it is too conservative. However, since the DKW bound is tight ( ``The Tight Constant in the Dvoretzky–Kiefer–Wolfowitz Inequality''. Massart 1990) it cannot be improved without additional assumptions.
>
> Q3) Situations where the method fails. The effectiveness of NACP is closely tied to $\Delta$ (the finite size correction term). When the calibration set is too small, $\Delta$ becomes large. Tables 2 and 4 illustrate the correction term as a function of the number of classes and the size of the calibration set.

---

### Meta-Review · Area_Chair_xpuT · 2024-12-17

**Metareview:**

After the rebuttal, several concerns remained unaddressed. Specifically the assumptions of known noise distribution are restrictive, and limit the applicability and interest in the work. While the authors motivate known noise with settings such as differential privacy, they do not present concrete theoretical or empirical results with DP. Further, the authors do not have theoretical results to show that estimating the noise distribution retains coverage; only showing this experimentally in certain cases (and it is unclear how general these are). Also, for the experiments, they only use synthetic examples where they add noise artificially, and no realistic examples where noise occurs naturally, which the reviewers found limits the motivation of the work.  Unfortunately, due to these issues, I feel that this paper needs more work. I recommend authors to address all issues for a future submission.

**Additional Comments On Reviewer Discussion:**

As mentioned above, after the rebuttal, several concerns remained: Specifically the assumptions of known noise distribution are restrictive, and limit the applicability and interest in the work. While the authors motivate known noise with settings such as differential privacy, they do not present concrete theoretical or empirical results with DP. Further, the authors do not have theoretical results to show that estimating the noise distribution retains coverage; only showing this experimentally in certain cases (and it is unclear how general these are). Also, for the experiments, they only use synthetic examples where they add noise artificially, and no realistic examples where noise occurs naturally, which the reviewers found limits the motivation of the work.

---

### Decision · Program_Chairs · 2025-01-22

Reject